# Systemic Changes in Endocannabinoids and Endocannabinoid-like Molecules in Response to Partial Nephrectomy-Induced Ischemia in Humans

**DOI:** 10.3390/ijms24044216

**Published:** 2023-02-20

**Authors:** Ariel Rothner, Tom Gov, Liad Hinden, Alina Nemirovski, Joseph Tam, Barak Rosenzweig

**Affiliations:** 1Obesity and Metabolism Laboratory, Institute for Drug Research, School of Pharmacy, Faculty of Medicine, The Hebrew University of Jerusalem, Jerusalem 9112001, Israel; 2Sackler Faculty of Medicine, Tel Aviv University, Tel Aviv 6905904, Israel; 3Department of Urology, Chaim Sheba Medical Center, Ramat Gan 5265601, Israel

**Keywords:** endocannabinoids, *N*-acylethanolamines, partial nephrectomy, renal ischemia reperfusion

## Abstract

Renal ischemia–reperfusion (IR), a routine feature of partial nephrectomy (PN), can contribute to the development of acute kidney injury (AKI). Rodent studies show that the endocannabinoid system (ECS) is a major regulator of renal hemodynamics and IR injury; however, its clinical relevance remains to be established. Here, we assessed the clinical changes in systemic endocannabinoid (eCB) levels induced by surgical renal IR. Sixteen patients undergoing on-clamp PN were included, with blood samples taken before renal ischemia, after 10 min of ischemia time, and 10 min following blood reperfusion. Kidney function parameters (serum creatinine (sCr), blood urea nitrogen (BUN), and serum glucose) and eCB levels were measured. Baseline levels and individual changes in response to IR were analyzed and correlation analyses were performed. The baseline levels of eCB 2-arachidonoylglycerol (2-AG) were positively correlated with kidney dysfunction biomarkers. Unilateral renal ischemia increased BUN, sCr, and glucose, which remained elevated following renal reperfusion. Renal ischemia did not induce changes in eCB levels for all patients pooled together. Nevertheless, stratifying patients according to their body mass index (BMI) revealed a significant increase in *N*-acylethanolamines (anandamide, AEA; *N*-oleoylethanolamine, OEA; and *N*-palmitoylethanolamine, PEA) in the non-obese patients. No significant changes were found in obese patients who had higher *N*-acylethanolamines baseline levels, positively correlated with BMI, and more cases of post-surgery AKI. With the inefficiency of ‘traditional’ IR-injury ‘preventive drugs’, our data support future research on the role of the ECS and its manipulation in renal IR.

## 1. Introduction

Partial nephrectomy (PN) has become the standard treatment for localized small renal masses, aiming to preserve renal function [1]. In most cases, PN relies on renal artery occlusion to improve visualization and reduce blood loss. However, the contribution of this transient ischemia and the subsequent reperfusion to renal outcomes has been highly debated [2]. Although lasting dogma held that an extended ischemia time (≥25 min) is associated with deleterious renal consequences [3], an intra-operative histological examination suggests that the human kidney has a higher tolerance to ischemia [4], with long-term renal outcomes supporting this claim [5]. Nevertheless, PN is associated with an acute decline in renal function, with an increase in serum creatinine (sCr) observed within the first few days post-surgery in single kidney patients [6]. At what point these markers increase and whether they respond to ischemia or reperfusion, in particular, remain unknown. A more in-depth understanding of the systemic response to ischemia–reperfusion (IR) may shed light on the variability in patients’ adverse responses following interventions.

With IR contributing to various cases of tissue damage, including myocardial infarction, stroke, and organ transplantation, its complex pathophysiology has been extensively researched. One of the regulators implicated in IR in various organs is the endocannabinoid system (ECS) [7]. The ECS is a network of G-protein-coupled receptors, the canonical cannabinoid-1 (CB_1_R) and cannabinoid-2 receptors (CB_2_R), their endogenous ligands (endocannabinoids or eCBs), most notably 2-arachidonoylglycerol (2-AG) and *N*-arachidonoylethanolamine (anandamide (AEA)), along with their synthesizing and catabolizing machinery. This network also comprises eCB-like compounds, *N*-oleoylethanolamine (OEA) and *N*-palmitoylethanolamine (PEA), which share their synthetic and metabolic pathways with AEA [8]. Both cannabinoid receptors have been localized to different compartments of the nephron, thus contributing to renal hemodynamics and function in a healthy physiological state. Additionally, alterations in the ECS contribute to the development of various renal pathologies, including chronic kidney disease (CKD) induced by diabetes or obesity and different forms of acute kidney injury (AKI) [9]. The involvement of the ECS in renal IR, in particular, has been demonstrated in rodent models, where renal arterial clamping induces changes in the kidney levels of 2-AG [10,11] and AEA [12] and in the expression of CB_1_R [12] and CB_2_R [13]. Furthermore, pre-clinical treatments with ECS-targeted therapies have been effective in preventing IR-induced kidney damage [10,11]. However, these studies were limited to rodent models, and their relevance to renal IR in humans remains to be established. 

This study aims to evaluate the systemic response of the ECS in settings of IR in patients undergoing PN. Herein, we report the dynamic intra-operative changes in kidney function markers and circulating eCBs before intervention, during ischemia, and 10-minute post-renal reperfusion.

## 2. Results

### 2.1. Renal Function and Surgical Data

The baseline characteristics, renal function, and surgical details of the study participants are presented in Table 1. The study included sixteen patients undergoing PN to excise renal tumors. One study participant had two tumors (2 + 1.1 cm), whereas all others had a single mass. All study subjects had an intact contralateral kidney. Pathologic diagnosis revealed that most of the patients with renal cell carcinoma (RCC) (56%), determined to be a clear cell RCC, and 25% were diagnosed with Papillary RCC. Surgical techniques included robotic intraperitoneal (n = 11), robotic extraperitoneal (n = 4), and open surgery (n = 1). During PN, the median ischemia time was 19 min (8–32 min), with a median estimated blood loss of 35 mL (0–700 mL). Within 48 h post-surgery, three patients (19%) experienced an episode of AKI according to the Kidney Disease: Improving Global Outcomes (KDIGO) clinical practice guidelines, with an increase in sCr by >0.3 mg/dL (Appendix A, Table A1). At the most recent follow-up (3 weeks to 6 months post-operation), the median individual changes in sCr and estimated glomerular filtration rate (eGFR) from pre-operative values were 0.06 mg/dL (−0.6–0.44) and −5 mL/min per 1.73 m^2^ (−41–16), respectively (Table A1 and Table A2).

### 2.2. Baseline Kidney Function Correlates with Higher 2-AG Levels

The circulating baseline levels of the main eCBs (2-AG and AEA), their related endogenous molecules (OEA and PEA), and the cumulative degradation product (arachidonic acid, AA) of 2-AG and AEA were measured (Table A3). A direct relationship was observed between the systemic 2-AG levels and kidney dysfunction parameters, sCr (r = 0.509, *p* = 0.044) and BUN (r = 0.548, *p* = 0.028), measured immediately before renal artery occlusion (pre-clamp) (Figure 1a,b). These baseline 2-AG levels also significantly corresponded to the pre-operative kidney dysfunction parameters, with 2-AG correlating positively to sCr (r = 0.540, *p* = 0.031) and negatively to eGFR (r = −0.573, *p* = 0.020) (Figure 1c,d). A positive correlation between 2-AG levels and tumor size was also observed, although it did not reach statistical significance (r = 0.510, *p* = 0.052; Figure 1e).

### 2.3. Unilateral Ischemia Increases the Circulating Kidney Dysfunction Parameters

To assess the effect of unilateral renal IR on the kidney function parameters, blood samples were collected pre-renal clamping (pre-clamp), 10 min post-renal clamping (ischemia), and 10 min following unclamping (reperfusion). Renal ischemia markedly increased BUN (1.5-fold [1.4–1.8]), serum glucose (1.4-fold [1.1–2.0]), and to a lesser extent, sCr (1.1-fold [1.0–1.3]) (Figure 2). These markers remained elevated in the circulation 10 minutes following reperfusion. Despite these increases, the 24 h post-operative measurements of sCr and serum glucose did not significantly differ from the pre-operative levels (*p* = 0.06 and *p* = 0.28, respectively) (Table A1 and Table A4).

### 2.4. Ischemia Increases Circulating N-Acylethanolamines in Non-Obese Patients

No significant changes were observed in eCBs, eCB-like molecules or AA due to ischemia or reperfusion for all 16 patients (Figure 3). Notably, the baseline levels of *N*-acylethanolamines (AEA, OEA, and PEA) were higher in the obese patients, positively correlated with their body mass index (BMI) (Figure 4). To account for these higher initial levels, subjects were stratified by BMI (non-obese (BMI < 30, n = 12) vs. obese (BMI > 30, n = 4)). In the non-obese patients, renal ischemia significantly increased the circulating *N*-acylethanolamines (AEA, 1.3-fold [0.8–2.0] *p* = 0.02; OEA, 1.2-fold [0.7–1.4], *p* = 0.02; PEA, 1.3-fold [0.6–1.8] *p* = 0.07) (Figure 5). In contrast, the obese patients displayed no eCB response to the renal occlusion (Figure 5). For all cases, the renal reperfusion showed varying responses in eCB levels, with no significant differences in comparison to the pre-clamp or ischemic state. Notably, two out of the three patients that experienced post-operative AKI were obese (Table A1). A third obese patient recorded an increase in sCr levels of 0.27 mg/dL. The non-obese patient that experienced AKI post-surgery was the only study participant to undergo an open surgery. 

## 3. Discussion

Ischemia and reperfusion to an organ causes a complex physiological response, which often leads to a pathological condition. In the ischemic state, the restriction of blood flow disturbs the metabolic supply and demand and leads to tissue hypoxia. Though oxygen is restored with blood reperfusion, an increase in reactive oxygen species, inflammatory response, and cell death programs exacerbate the tissue damage [14]. Research on renal IR is abundant; however, the dissonance between animal models and humans is a major pitfall. The extensive renal damage and functional decline, observed immediately in rodent models, are not completely imitated in humans following surgical renal artery occlusion. Furthermore, various biomarker studies and therapeutic interventions in preclinical IR failed to replicate their effectiveness in humans [15]. Moreover, former ‘traditional’ human kidney IR injury prevention drugs were found to be inefficient [16]. Consequently, a thorough understanding of the human pathophysiological response to renal IR and its contribution to AKI is needed.

In this study, we found that kidney injury, in the form of sCr elevation, occurs concomitant with renal clamping in patients undergoing PN. The other circulating metabolites, BUN and serum glucose, were also immediately increased due to renal ischemia, more drastically than was sCr. Although these levels remained elevated after renal reperfusion, they normalized by the first day post operation, probably representing both kidney recovery, as well as compensation by the healthy contra-lateral kidney. In the general population, an AKI incidence is a high-risk factor for CKD development. Although this association is debatable for AKI following PN [17,18], we believe that elucidating the mechanism underlying surgical IR-induced AKI may shed light on the kidney injury and recovery processes.

This study is the first to evaluate the ECS response to renal IR in humans. By self-comparison at multiple time points, the study design offsets the multiple physiological factors known to affect circulating eCBs, and it provides a complete picture of the dynamic eCB changes per individual [19]. The ischemia-induced elevation in AEA levels we observed in non-obese patients provides clinical relevance to ECS involvement reported by numerous preclinical studies. AEA mediates renal hemodynamics by relaxing the vasculature of the renal endothelium [20], dilating afferent and efferent arterioles, and increasing renal blood flow [21]. Furthermore, rodent models of IR-induced injury have displayed decreases in kidney AEA levels [10,12]. Like AEA, the eCB-like compounds, OEA and PEA, were systemically elevated in response to ischemia. These three *N*-acylethanolamine compounds share common metabolic pathways, suggesting an involvement in their synthetic or degradative enzymes [8]. Indeed, fatty acid amide hydrolase (FAAH), the primary enzyme responsible for *N*-acylethanolamine degradation, was overexpressed in a mouse model post-renal IR injury [22]. Rodent studies support a protective role of *N*-acylethanolamine in IR injury. PEA administration attenuated the inflammation, injury, and renal functional decline when administered concurrently with renal ischemia [23]. Moreover, in Chen et al. 2022, increasing renal AEA by FAAH inactivation or transgenic knockout mice was found to mediate renal fibrogenesis observed post-IR injury [22]. Though this study did not measure other *N*-acylethanolamines, others have demonstrated that a transgenic mouse strain lacking the FAAH enzyme (FAAH^−/−^) reveals a significant elevation in the levels of AEA, OEA, and PEA [24]. In light of these pre-clinical studies, our findings may suggest that the overall increase in circulating *N*-acylethanolamines 10 min following ischemia represents an immediate way for the body to protect itself from the detrimental consequences of artery clamping. However, larger and long-term follow-up studies are needed to determine whether the *N*-acylethanolamine elevation we observed in human renal ischemia indeed serves as a protective function. 

Notably, only the non-obese patients displayed ischemia-induced elevation in *N*-acylethanolamines, with no response observed in the four patients with BMI > 30. In line with the literature [25], we observed that baseline AEA, OEA, and PEA levels directly correlated with the patients’ BMI. This initial hyper *N*-acylethanolamine ‘tone’ in the obese patients may be implicated for their lack of response to renal ischemia. Whether this absence of eCB response is connected to the post-operative increases in sCr observed in three out of four of these patients needs further investigation. This research is especially important, since obesity is associated with a higher incidence of RCC [26]. 

Elucidating the role of *N*-acylethanolamines in renal ischemia may provide a therapeutic opportunity to mitigate the damage caused by renal IR. Further examination of the functional role of the ECS in this pathological clinical setting, including the expression and localization of the cannabinoid receptors and eCB metabolic enzymes, is needed. ECS homeostasis is altered by the ingestion of cannabis, which contains phytocannabinoids that compete with the canonical receptors [27]. This study supports further investigation of novel ECS-targeted drugs or exploiting phytocannabinoid-induced ECS changes as preventative therapies in various clinical settings that result in renal IR. 

Whereas systemic 2-AG levels showed no changes throughout the PN procedure, the levels of this eCB were higher in patients with lower kidney function (significantly correlated with sCr, BUN, and eGFR) and larger tumor masses. The ECS is known to regulate cancer proliferation, migration, and tumorigenesis, with its impact depending on the cancer type and organ [28]; however, its role in RCC is not well established. Early studies found a down-regulation of CB_1_R and no expression of CB_2_R in clear cell RCC human tissue in comparison with non-cancerous tissue [29]. In contrast, later studies in larger patient cohorts found CB_1_R and CB_2_R to be overexpressed in the RCC cancer tissue. For both receptors, higher expression levels in the cancer tissue were associated with reduced patient survival [30,31]. In this context, the association we found between tumor size/renal decline and systemic 2-AG, a full agonist of CB_1_R and CB_2_R, warrants further interest in the ECS mediation of the RCC progression, pointing to the rationale of considering systemic 2-AG as a biomarker for RCC. Moreover, the enhanced levels of systemic 2-AG may in themselves be a source of renal decline, with renal CB_1_R overactivity contributing to the pathogenesis of various kidney diseases [9]. Finally, considering the known ECS-modulation of both cancer progression and metabolic disorders, the link between RCC and obesity amidst the observed eCB changes found in this study is worth further investigation. 

The limitations of this study include its small population size and the lack of follow-up samples to understand the complete eCB profile changes. Additionally, this study focused on the systemic responses to renal IR, thus providing no data on the eCB changes within the occluded kidney. Since all patients had a functioning contralateral kidney, the question of whether the systemic eCB changes originate in the occluded kidney or they are a function of the compensatory mechanism of the non-affected kidney is unclear.

## 4. Materials and Methods

### 4.1. Study Population

Patients over 18 years of age undergoing PN to excise a renal tumor were included. Their medical histories regarding the demographic details and comorbidities were obtained from their medical records. The BMI (the body mass per height squared (kg/m^2^)), was calculated with a BMI > 30 considered obese.

### 4.2. Study Protocol

Data were collected after the patients’ written informed consent and a Sheba Medical Center institutional review board’s approval was obtained. Surgeries were performed in a single tertiary center from November 2021 to February 2022 by three urologic-oncology fellowship-trained surgeons with at least five years of experience as leading surgeons. After an overnight fast, patients underwent general anesthesia, and surgery was performed. Blood samples were collected before unilateral renal artery occlusion, after 10 min of ischemia time, and 10 min following reperfusion. 

### 4.3. Biochemistry Measurements

Pre- and post-operative levels of serum creatinine, serum glucose, and eGFR were collected from patients’ medical charts. Kidney function was calculated as the eGFR by means of the chronic kidney disease epidemiology collaboration (CKD-EPI) formula. The intra-operative blood samples (indicated as pre-clamp, ischemia, and reperfusion) were analyzed using a Cobas C-111 chemistry analyzer (Roche, Switzerland), and measured for serum urea, creatinine, and glucose. BUN was calculated by the serum urea levels (BUN mg/dL = Urea mM × 2.801).

### 4.4. Endocannabinoid Extraction and Measurement by LCMS/MS

The serum eCBs were extracted, purified, and quantified by stable isotope dilution liquid chromatography/tandem mass spectrometry (LC-MS/MS) as previously described [27]. In brief, serum proteins were first precipitated with ice-cold acetone and Tris buffer (50 mM, pH 8.0). Next, an ice-cold extraction buffer (1:1 MeOH/Tris Buffer + an internal standard (IS; d4-AEA)) were added to the samples. Homogenates were then extracted using ice-cold CHCl_3_:MeOH (2:1, *v*/*v*), and then washed with ice-cold chloroform three times. The samples were then dried under nitrogen and reconstituted in methanol. Analysis by LC-MS/MS was carried out on an AB Sciex (Framingham, MA, USA) QTRAP® 6500 + mass spectrometer coupled with a Shimadzu (Kyoto, Japan) UHPLC System. Liquid chromatographic separation was carried out using 5 μL injections of samples onto a Kinetex 2.6 µm C18 (100 × 2.1 mm) column from Phenomenex (Torrance, CA, USA). The autosampler was set at 4 °C and the column was maintained at 40 °C during the entire analysis. Gradient elution mobile phases consisted of 0.1% formic acid in water (Phase A) and 0.1% formic acid in acetonitrile (Phase B). eCBs were detected in a positive ion mode using electron spray ionization (ESI), and by the multiple reaction monitoring (MRM) mode of acquisition, using d4-AEA as an IS. The collision energy (CE), declustering potential (DP), and the collision cell exit potential (CXP) for the monitored transitions are presented in Table A5. The levels of 2-AG, AEA, OEA, PEA, and AA in the samples were measured against standard curves, and calculated for pmol/mL serum. 

### 4.5. Statistical Analysis

Statistical analysis was performed using GraphPad Prism version 9. Continuous variables are expressed as median (range) and categorical variables as proportions. The Pearson correlation coefficient and univariate linear regression analysis were used to assess correlations between baseline eCB levels and continuous variables (eGFR, sCr, BUN, and BMI). To assess the effect of IR on eCB levels, accounting for the varying individual baseline levels, the fold change from pre-clamping was calculated for each patient. One-way repeated measure ANOVA with Greenhouse–Geisser correction was performed, with a post hoc Tukey multiple comparison test, to compare the intra-operative groups. Pre-operative and post-operative biochemistry values were compared using a paired Student’s *t*-test. *p* < 0.05 was considered statistically significant for all analyses.

## 5. Conclusions

This study revealed a temporal dynamic pattern of kidney dysfunction markers in response to surgical renal IR, elucidating just how immediately the levels are changed. Renal occlusion increased circulating *N*-acylethanolamine levels in non-obese patients, thus providing clinical evidence for ECS involvement in renal IR. In contrast, obese patients, who had a higher eCB tone, displayed no ischemia-induced elevation in *N*-acylethanolamines, along with a worse post-operative acute renal decline. How this increase in IR by *N*-acylethanolamines can influence long-term renal recovery in different patient populations and extensive investigation of the functional role of these and other ECS changes requires further research. With the diverse clinical scenarios of renal IR and the current lack of preventive drug interventions, ECS targeting provides a novel therapeutic opportunity.

## Figures and Tables

**Figure 1 ijms-24-04216-f001:**
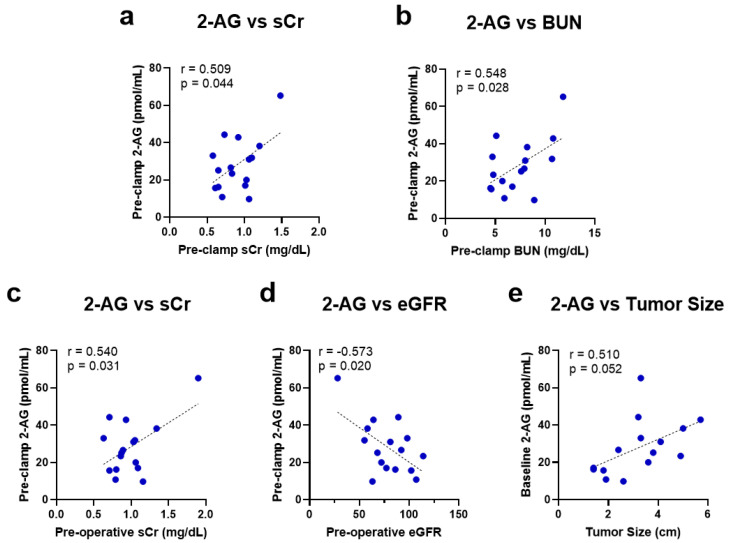
Univariate linear regression analysis and the Pearson correlation coefficient of baseline 2-AG levels with intra-operative pre-renal artery clamping levels of (**a**) sCr and (**b**) BUN and pre-operative levels of (**c**) sCr, (**d**) eGFR, and (**e**) tumor size. 2-AG = 2-arachidonoylglycerol; sCr = serum creatinine; BUN = blood urea nitrogen; eGFR = estimated glomerular filtration rate.

**Figure 2 ijms-24-04216-f002:**
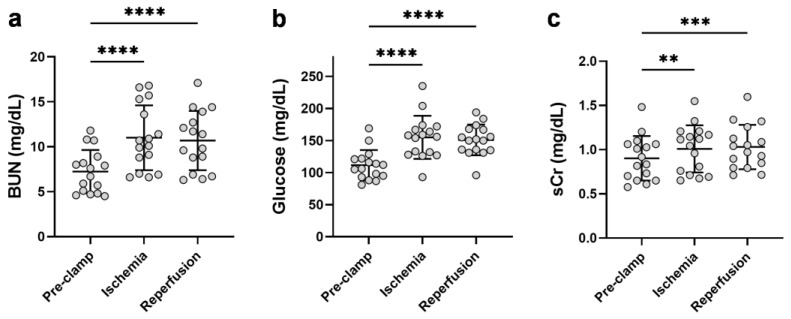
Effect of renal ischemia and reperfusion during partial nephrectomy on the circulating kidney dysfunction markers, (**a**) BUN, (**b**) glucose, and (**c**) sCr. BUN = blood urea nitrogen; sCr = serum creatinine. **** *p* < 0.0001; *** *p* < 0.001; ** *p* < 0.01.

**Figure 3 ijms-24-04216-f003:**
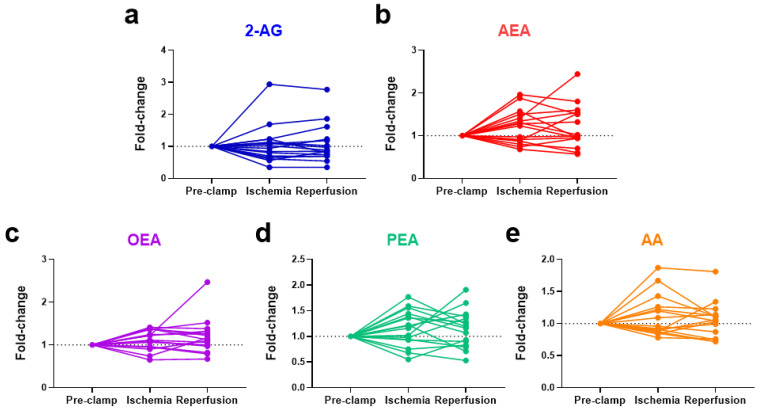
Changes in the systemic levels of endocannabinoids, ((**a**) 2-AG and (**b**) AEA), endocannabinoid-like compounds, ((**c**) OEA and (**d**) PEA), and a degradative product ((**e**) AA) as a result of renal ischemia and reperfusion during partial nephrectomy. 2-AG = 2-arachidonoylglycerol; AEA = anandamide; OEA = *N*-oleoylethanolamine; PEA = *N*-palmitoylethanolamine; AA = arachidonic acid.

**Figure 4 ijms-24-04216-f004:**
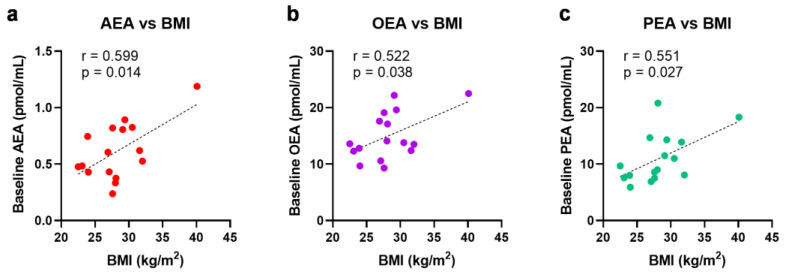
Univariate linear regression analysis and the Pearson correlation coefficient of BMI with baseline *N*-acylethanolamine levels, (**a**) AEA, (**b**) OEA, and (**c**) PEA. BMI = body mass index; AEA = anandamide; OEA = *N*-oleoylethanolamine; PEA = *N*-palmitoylethanolamine.

**Figure 5 ijms-24-04216-f005:**
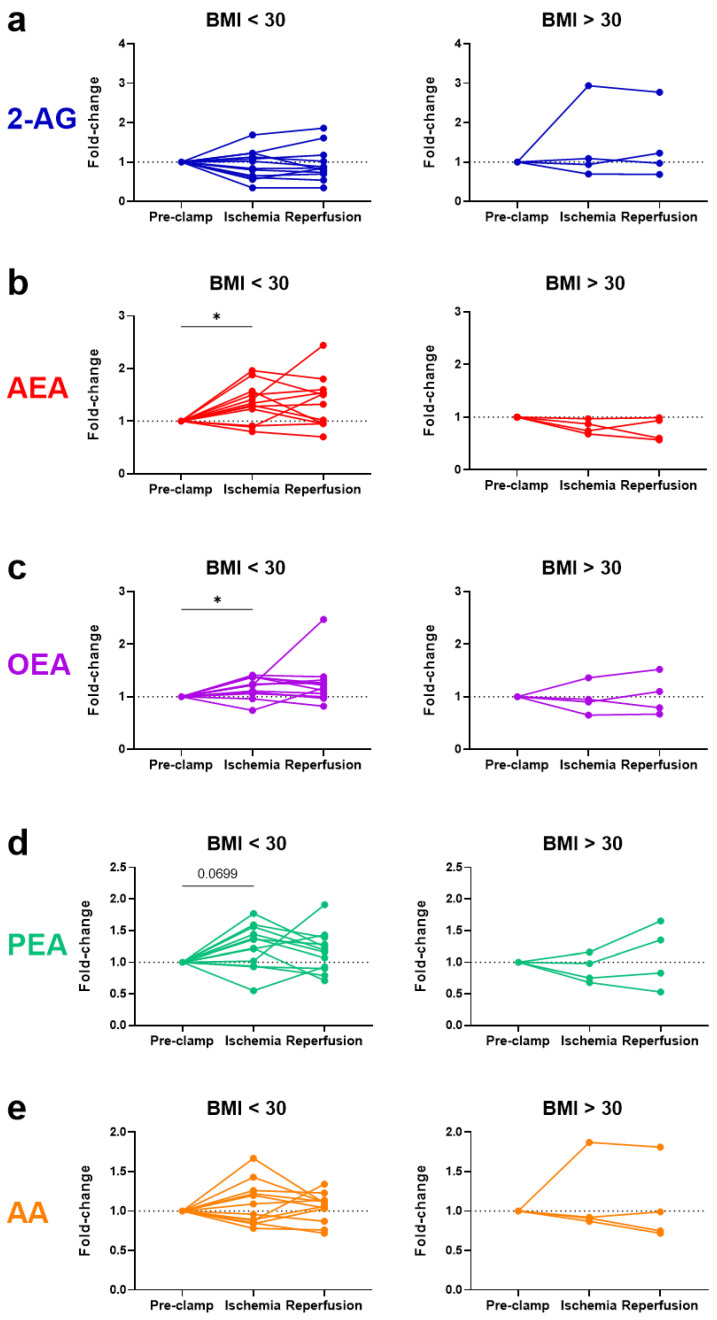
Patient stratification by BMI displaying changes in the systemic levels of endocannabinoids, ((**a**) 2-AG and (**b**) AEA), endocannabinoid-like compounds, ((**c**) OEA and (**d**) PEA), and a degradative product ((**e**) AA) as a result of renal ischemia–reperfusion during partial nephrectomy. Results are stratified for non-obese (BMI < 30; left panels) versus obese (BMI > 30; right panels) patients. BMI = body mass index; 2-AG = 2-arachidonoylglycerol; AEA = anandamide; OEA = *N*-oleoylethanolamine; PEA = *N*-palmitoylethanolamine; AA = arachidonic acid. * *p* < 0.05.

**Table 1 ijms-24-04216-t001:** Demographic, surgical, and renal functional data at baseline.

Parameters	n = 16
*Demographic Data*	
Age (years)	68 (37–79)
Women, n (%)	8 (50)
BMI (kg/m^2^)	27.8 (22.5–40.1)
Hypertension, n (%)	7 (44)
Diabetes mellitus, n (%)	4 (25)
*Renal Functional and Tumor Data*	
Creatinine (serum) (mg/dL)	0.91 (0.63–1.90)
eGFR (mL/min per 1.73 m^2^)	79 (28–114)
BUN (mg/dL)	7.2 (4.5–11.8)
Tumor Size ^1^ (cm)	3.3 (1.4–5.7)
RCC Pathology, n (%)	
Clear cell RCC	9 (56)
Papillary RCC	4 (25)
Oncocytoma	3 (19)
*Surgical Data*	
Surgical Technique	
Robotic intraperitoneal	11 (69)
Robotic extraperitoneal	4 (25)
Open	1 (6)
Ischemia Time (min)	19 (8–32)

BMI—body mass index; eGFR—estimated glomerular filtration rate; BUN—blood urea nitrogen; RCC—renal cell carcinoma. Data are expressed as the median (range) for continuous values and as a number (percent) for categorical values. ^1^ Tumor size based on pre-operative imaging.

## Data Availability

The entire data set presented in this study is available within the article and Appendix A.

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
