# Peer review of "Systemic Changes in Endocannabinoids and Endocannabinoid-like Molecules in Response to Partial Nephrectomy-Induced Ischemia in Humans"

_ijms, 2023, doi:10.3390/ijms24044216_

Round 1

Reviewer 1 Report

In this study, Rothner et al., have addressed how systemic endocannabinoid levels are modulated in response to surgical renal IR by correlating this modulation with classical kidney function parameters and surgical data. Endocannabinoid tone has been widely described as a major regulator of renal hemodynamic and IR injury in animal models. However, its clinical relevance has not determined yet.

By using human samples, the investigators evaluated the dynamic (before-intra-after intervention) and systemic response of endocannabinoids in settings of ischemia-riperfusion in patients undergoing PN. They found that one N-acylethanolamine (2-AG) positively correlated with serum creatine and blood urea nitrogen, and that AEA and OEA were significantly higher after renal ischemia only in non-obese subjects.

The statistical power is not indicated, did the author calculate it to confirm that n=16 is not too small as population size? The work would have benefit from additional analyses. In particular, it would have been interesting to see how long-lasting the effect on EC tone after reperfusion is. Moreover, circulating levels of endocannabinoids are not indicative of changes in the local injured kidney microenvironment and do not allow to understand whether the systemic changes reported are instead due to a compensatory response of the non-affected kidney.

Despite the above-mentioned limitations, which indeed were disclosed and commented by the authors; these interesting findings provide a novel concept and data shed light into a possible protective role of ECs in clinical KD.

Previous studies reported an increased EC tone in the early phase of several (non-renal) tumors as a possible response to counteract carcinogenesis. Thus, considering the higher incidence of renal carcinoma in obese/diabetic people, the differences observed in the EC levels with respect to obese phenotype of patients are worth to be further investigated.

Overall, the paper provides novel and interesting data, is well written, and the current data support the ideas offered in the paper. I can only suggest minor revisions to implement the story telling of the manuscript. While reading the manuscript, I noticed that the stratification of the samples is already reported in Figure 3, before the authors describe the incremented baseline levels of N-acylethanolamines in patients with BMI>30. I suggest inverting the order of figures and/or re-organize figures by first showing the positive correlation of Figure 4 and later the BMI-clustered data (see 2nd and 3rd columns in Figure 3).

Finally, as a conclusive statement I suggest stressing more the urgent need of further investigations characterizing the functional role of ECS (i.e. CBR expression and/or localization, EC-related enzyme expression/activity) in such pathological clinical settings rather than claiming the targeting of ECS as a novel therapeutic opportunity. 

Author Response

Reviewer 1:

In this study, Rothner et al., have addressed how systemic endocannabinoid levels are modulated in response to surgical renal IR by correlating this modulation with classical kidney function parameters and surgical data. Endocannabinoid tone has been widely described as a major regulator of renal hemodynamic and IR injury in animal models. However, its clinical relevance has not determined yet.

By using human samples, the investigators evaluated the dynamic (before-intra-after intervention) and systemic response of endocannabinoids in settings of ischemia-riperfusion in patients undergoing PN. They found that one N-acylethanolamine (2-AG) positively correlated with serum creatine and blood urea nitrogen, and that AEA and OEA were significantly higher after renal ischemia only in non-obese subjects.

The statistical power is not indicated, did the author calculate it to confirm that n=16 is not too small as population size? The work would have benefit from additional analyses.

As mentioned in the discussion section, the small sample size was a limitation of the study. As per your review, we performed a post-hoc power analysis using the G*Power 3 program, and found the study to be underpowered (<80%) (Faul, F., Erdfelder, E., Lang, A.-G., & Buchner, A. (2007). G*Power 3: A flexible statistical power analysis program for the social, behavioral, and biomedical sciences. Behavior Research Methods39, 175-191). Nevertheless, we still found that the renal ischemia induced statistically significant changes in AEA and OEA in the non-obese patients. It also worth mentioning, as detailed in the Statistical Analysis sub section, that in order to account for the varying individual baseline endocannabinoid levels, the fold change from pre-clamping was calculated for each patient (which strengthen the statistical power by itself), and one-way repeated measure ANOVA with Greenhouse-Geisser correction was performed, with a post-hoc Tukey multiple comparison test, to compare the intra-operative groups.  

In particular, it would have been interesting to see how long-lasting the effect on EC tone after reperfusion is. Moreover, circulating levels of endocannabinoids are not indicative of changes in the local injured kidney microenvironment and do not allow to understand whether the systemic changes reported are instead due to a compensatory response of the non-affected kidney.

We agree with these comments, which were specifically addressed in the last paragraph in the discussion. 

Despite the above-mentioned limitations, which indeed were disclosed and commented by the authors; these interesting findings provide a novel concept and data shed light into a possible protective role of ECs in clinical KD.

We would like to thank the reviewer for finding our manuscript interesting and novel.

Previous studies reported an increased EC tone in the early phase of several (non-renal) tumors as a possible response to counteract carcinogenesis. Thus, considering the higher incidence of renal carcinoma in obese/diabetic people, the differences observed in the EC levels with respect to obese phenotype of patients are worth to be further investigated.

This is an extremely relevant point to both the findings of our study and the on-going research of ECS-modulation in carcinogenesis and metabolic disorders. As such, we have added a sentence highlighting this idea (See Page 8, lines 247-249).

Overall, the paper provides novel and interesting data, is well written, and the current data support the ideas offered in the paper. I can only suggest minor revisions to implement the story telling of the manuscript. While reading the manuscript, I noticed that the stratification of the samples is already reported in Figure 3, before the authors describe the incremented baseline levels of N-acylethanolamines in patients with BMI>30. I suggest inverting the order of figures and/or re-organize figures by first showing the positive correlation of Figure 4 and later the BMI-clustered data (see 2nd and 3rd columns in Figure 3).

Again, we would like to deeply thank the reviewer for finding our manuscript interesting and novel. As per the suggestion raised by the reviewer, we have split Figure 3 to two separate Figures. The New Figure 3 presents the non-stratified data for all patients and the New Figure 5 presents the data stratified by BMI. The text was amended accordingly. We hope this re-organization better clarifies the story-telling of the paper.

Finally, as a conclusive statement I suggest stressing more the urgent need of further investigations characterizing the functional role of ECS (i.e. CBR expression and/or localization, EC-related enzyme expression/activity) in such pathological clinical settings rather than claiming the targeting of ECS as a novel therapeutic opportunity. 

We agree that a full characterization of the ECS in this setting would contribute to the research, and have added text emphasizing this need in the discussion section (See Page 8, lines 226-228) and conclusion section (See Page 9, line 319).

Reviewer 2 Report

The manuscript named “Systemic Changes in Endocannabinoids in Response to Partial Nephrectomy-induced Ischemia” is dedicated to the important clinical area of ischemia-reperfusion changes during nephrectomy systemic alteration of endocannabinoids and non-endocannabinoid N-acylethanolamines. Better understanding of these processes, their mechanisms, regulation, and systemic effects will potentially allow to develop more effective approach for the treatment of kidney injury and improve outcome of the nephrectomy. However, to enhance the quality of the paper some improvement and clarification of data presentation, discussion, and conclusion have to be made.

 1). The title of the paper has to be edited because PEA, OEA are not endocannabinoids because they do not produce effects through CB1/2 receptors. However, changes of these N-acylethanolamines are studies in the manuscript.

2). All abbreviations must be explained in the places of the paper where they first appeared. One of example is the BMI, which was in the abstract but transcribed only in Table 1.

 3). Line 54. “This network also comprises eCB-like compounds, N-oleoylethanolamine (OEA) and N-palmitoylethanolamine (PEA), which share their catabolic pathway with AEA” PEA shares with AEA metabolic but not catabolic pathways. This should be also considered in the discussion.

 4). Line 92. 2.2. Baseline Kidney Function Correlates with Higher 2-AG Levels.  According to the study protocol (line 234) blood collection has been performed after premedication and general anesthesia just before renal artery occlusion. Therefore, at this point kidney functions could not be considered as that of basal conditions. They could be affected by general anesthesia and surgical manipulations. These functions are reflected conditions before the occlusion. These should be considered in the presentation of results and in the discussion.

 5). Line 120. “No significant changes were observed in eCB levels due to ischemia or reperfusion for all 16 patients (Figure 3). The baseline levels of N-acylethanolamines (AEA, OEA, and PEA) were higher in the obese patients, positively correlated with their BMI (Figure 4).” AEA (but not OEA and PEA) is also eCB that changes due to ischemia. Description of obtained results should be edited.  

 6). Line 162. “Although these levels remained elevated after renal reperfusion, they normalized by 1-day post-operation, probably representing..” The conclusion about normalization of parameters has to be supported by statistical data.

  7). Line 180. “Indeed, fatty acid amide hydrolase (FAAH), the primary enzyme responsible for N-acylethanolamine degradation, was overexpressed in a mouse model post-renal IR injury [22].” FAAH is primary responsible for degradation of AEA only but not all N-acylethanolamines. In the cited work only AEA has been studied. 

Author Response

The manuscript named “Systemic Changes in Endocannabinoids in Response to Partial Nephrectomy-induced Ischemia” is dedicated to the important clinical area of ischemia-reperfusion changes during nephrectomy systemic alteration of endocannabinoids and non-endocannabinoid N-acylethanolamines. Better understanding of these processes, their mechanisms, regulation, and systemic effects will potentially allow to develop more effective approach for the treatment of kidney injury and improve outcome of the nephrectomy. However, to enhance the quality of the paper some improvement and clarification of data presentation, discussion, and conclusion have to be made.

We would like to thank the reviewer for highlighting the importance of our study. We have revised the manuscript according to his/her comments/suggestions.

 1). The title of the paper has to be edited because PEA, OEA are not endocannabinoids because they do not produce effects through CB1/2 receptors. However, changes of these N-acylethanolamines are studies in the manuscript.

This is a very valid point, and we therefore have amended the title to include endocannabinoid-like molecules as well (See Page 1, lines 1-3).

2). All abbreviations must be explained in the places of the paper where they first appeared. One of example is the BMI, which was in the abstract but transcribed only in Table 1.

The paper was reviewed with this in mind, and abbreviations were correctly defined. See Page 1, line 26; Page 2, line 89; Page 5, line 140.

3). Line 54. “This network also comprises eCB-like compounds, N-oleoylethanolamine (OEA) and N-palmitoylethanolamine (PEA), which share their catabolic pathway with AEA” PEA shares with AEA metabolic but not catabolic pathways. This should be also considered in the discussion.

As described in Tsuboi et. al 2018, the N-acylethanolamines, including AEA, OEA, and PEA, share both synthetic and metabolic pathways. As such, we have edited the text to reflect this point (See Page 2, line 59). We further highlighted the role of their shared metabolic pathway in the discussion, as per comment #7 below.

4). Line 92. 2.2. Baseline Kidney Function Correlates with Higher 2-AG Levels.  According to the study protocol (line 234) blood collection has been performed after premedication and general anesthesia just before renal artery occlusion. Therefore, at this point kidney functions could not be considered as that of basal conditions. They could be affected by general anesthesia and surgical manipulations. These functions are reflected conditions before the occlusion. These should be considered in the presentation of results and in the discussion.

Thank you for a very compelling point. The intra-operative blood samples taken right before renal artery occlusion (pre-clamp) were used to assess both kidney dysfunction markers (serum creatinine and BUN) and eCB levels. As such, the correlation found between the pre-clamp 2-AG levels and the high kidney dysfunction levels could indeed be affected by the general anesthesia and surgical manipulations. We performed additional correlation analyses on these pre-clamp 2-AG levels with the pre-operative (i.e. before any surgical intervention) kidney dysfunction markers (serum creatinine and eGFR), taken from the patients’ medical records- within 24-hours of surgery. Interestingly, the significant correlations between baseline 2-AG levels and kidney dysfunction markers were also found with these pre-operative levels. As per the comments, we have decided to include these additional correlation analyses (See Page 3, lines 109-113, and Figure 1) and clarified in the method section the different parameters used for comparisons (See Page 9, line 276).

5). Line 120. “No significant changes were observed in eCB levels due to ischemia or reperfusion for all 16 patients (Figure 3). The baseline levels of N-acylethanolamines (AEA, OEA, and PEA) were higher in the obese patients, positively correlated with their BMI (Figure 4).” AEA (but not OEA and PEA) is also eCB that changes due to ischemia. Description of obtained results should be edited.  

The sentence was edited for clarification, See Page 5, line 137.

6). Line 162. “Although these levels remained elevated after renal reperfusion, they normalized by 1-day post-operation, probably representing.” The conclusion about normalization of parameters has to be supported by statistical data.

Statistical analysis of the pre-operative versus post-operative levels of serum creatinine and glucose was performed, and found to be not statistically significant. The p values from these tests were added to the results section (See Page 4, line 131). In addition, the test description was added to the statistical analysis section (See Page 9, lines 308-310).

7). Line 180. “Indeed, fatty acid amide hydrolase (FAAH), the primary enzyme responsible for N-acylethanolamine degradation, was overexpressed in a mouse model post-renal IR injury [22].” FAAH is primary responsible for degradation of AEA only but not all N-acylethanolamines. In the cited work only AEA has been studied. 

As described by Tsuboi et. al 2018, though FAAH is primarily responsible for AEA degradation, it has been found to hydrolyze various N-acylethanolamines. While the paper cited in our manuscript (Chen, C 2022) only measured the effect of the FAAH knockout on the levels of AEA, others have shown that a transgenic mouse strain lacking the FAAH enzyme (FAAH-/-) displays significant elevation in the levels of AEA, OEA, and PEA (Cravatt et. al PNAS 2001). We have included a statement elaborating this point in the discussion section (See Page 7, lines 207-212).
